



# Tropospheric observations of CFC-114 and CFC-114a with a focus on long-term trends and emissions

Johannes C. Laube[1], Norfazrin Mohd Hanif[1], Patricia Martinerie[2], Eileen Gallacher[1], Paul J. Fraser[3], Ray Langenfelds[3], Carl A. M. Brenninkmeijer[4], Jakob Schwander[5], Emmanuel Witrant[6], Jia-Lin Wang[7], Chang-Feng Ou-Yang[8], Lauren J. Gooch[1], Claire E. Reeves[1], William T. Sturges[1] and David E. Oram[1,9]

[1]Centre for Ocean and Atmospheric Sciences, School of Environmental Sciences, University of East Anglia, Norwich, NR4 7TJ, United Kingdom
[2]UJF-Grenoble 1/CNRS, Laboratoire de Glaciologie et Géophysique de l'Environnement, F-38041, Grenoble, France
[3]Oceans and Atmosphere, Commonwealth Scientific and Industrial Research Organisation, Aspendale, Australia
[4]Air Chemistry Division, Max Planck Institute for Chemistry, Mainz, Germany
[5]Physics Institute, University of Berne, Bern, Switzerland
[6]UJF-Grenoble 1/CNRS, Grenoble Image Parole Signal Automatique, Grenoble, France
[7]Department of Chemistry, National Central University, Zhongli, Taiwan
[8]Department of Atmospheric Sciences, National Central University, Taiwan
[9]National Centre for Atmospheric Science, School of Environmental Sciences, University of East Anglia, Norwich, NR4 7TJ, United Kingdom

*Correspondence to*: Johannes C. Laube (j.laube@uea.ac.uk)

**Abstract.** Chlorofluorocarbons (CFCs) are ozone depleting substances as well as strong greenhouse gases, and the control of their production and use under the Montreal Protocol has had demonstrable benefits to both mitigation of increasing surface UV radiation and climate forcing. A global ban on consumption came into force in 2010, but there is evidence of continuing emissions of certain CFCs from a range of sources. One compound has received little attention in the literature, namely CFC-114 ($C_2Cl_2F_4$). Of particular interest here is the differentiation between CFC-114 ($CClF_2CClF_2$) and its asymmetric isomeric form CFC-114a ($CF_3CCl_2F$) as atmospheric long-term measurements in the peer-reviewed literature to date have been assumed to represent the sum of both isomers with a time-invariant isomeric speciation. Here we report the first long-term measurements of the two isomeric forms separately, and find that they have different origins and trends in the atmosphere.

Air samples collected at Cape Grim (41°S), Australia, during atmospheric background conditions since 1978, combined with samples collected from deep polar snow (firn) enable us to obtain a near-complete record of both gases since their initial production and release in the 1940s. Both isomers were present in the unpolluted atmosphere in comparably small amounts before 1960. The mixing ratio of CFC-114 doubled from 7.9 to 14.8 parts per trillion (ppt) between the start of the Cape Grim record in 1978 and the end of our record in 2014, while over the same time CFC-114a trebled from 0.35 to 1.03 ppt. Mixing ratios of both isomers are slowly decreasing by the end of this period. This is consistent with measurements of recent aircraft-based samples showing no significant interhemispheric mixing ratio gradient.

We also find that the fraction of CFC-114a mixing ratio relative to that of CFC-114 increased from 4.3% to 6.9% over the 37-year period. This contradicts the current tacit assumption used in international climate change and ozone depletion



assessments that both isomers have been largely co-emitted and that their atmospheric concentration ratio has remained approximately constant in time. Complementary observations of air collected in Taiwan indicate a persisting source of CFC-114a in South East Asia which may have been contributing to the changing balance between the two isomers.

In addition we present top-down global annual emission estimates of CFC-114 and CFC-114a derived from these
measurements using a two-dimensional atmospheric chemistry-transport model. In general, the emissions for both compounds grew steadily during the 1980s, followed by a substantial reduction from the late 1980s onwards, which is consistent with the reduction of emission in response to the Montreal Protocol, and broadly consistent with bottom-up estimates derived by industry. However, we find that small but significant emissions of both isomers remain in 2014. Moreover the inferred changes to the ratio of emissions of the two isomers since the 1990s also indicate that the sources of
the two gases are, in part, independent.

## 1 Introduction

Chlorofluorocarbons (CFCs) are halogenated hydrocarbons of exclusively anthropogenic origin. These relatively inert gases were widely used in a variety of applications (e.g. as refrigerants, propellants and foam-blowing agents) since the 1930s. CFCs tend to persist in the atmosphere due to the absence of a significant tropospheric sink process. They eventually are
broken down in the stratosphere thereby releasing reactive chlorine, which catalytically destroys ozone (Rowland and Molina, 1975), resulting in increases in harmful UV radiation at the surface. Since the discovery of the Antarctic ozone hole (Farman et al. 1985), the enforcement and subsequent amendments of the Montreal Protocol on Substances that Deplete the Ozone Layer (1989) resulted in the successful ban of CFC consumption (= production + imports – exports) in industrialised and developing nations by 2010 (apart from relatively minor critical-use exemptions). Consequently, the abundance of most
documented CFCs in the atmosphere started to reduce (Montzka et al., 1996a, Montzka and Reimann et al., 2010, Rigby et al. 2013, Carpenter and Reimann et al. 2014, Laube et al. 2014; for Montreal Protocol ratification status see UNEP, 2014).
Continued research and monitoring of all ozone depleting substances (ODSs) are essential to ensure compliance with the Montreal Protocol for environmental protection against ozone loss.

This study focuses on two CFC compounds that have been particularly understudied to date. The isomeric pair of CFC-114
($CClF_2CClF_2$) and asymmetric CFC-114a ($CF_3CCl_2F$) were primarily used as aerosol propellants, as blowing agents in polyolefin foams and as refrigerants in long-lived appliances (Fisher and Midgley, 1993) before production was banned following the Montreal Protocol (UNEP, 2014). Minor remaining uses of CFC-114 were for cooling processes e.g. in naval vessels (Andersen et al. 2007). CFC-114a has also been reportedly used in the production of HFC-134a, the latter being one of the alternatives to replace CFC-114 used in chillers (Banks et al. 1994). The Alternative Fluorocarbons Environmental
Acceptability Study (AFEAS) reported that 514,319 tons of the isomers (combined) were produced between 1937 and 2004, primarily in the Northern Hemisphere, ~15% of which was used within long-lived applications (>12 years lifetime),





providing a substantial bank of the isomers to potentially produce continued emissions to the atmosphere (AFEAS, 1995 and 2009).

As with all CFCs, the stratospheric loss processes of these isomers are photolysis by ultra-violet radiation and reaction with excited-state atomic oxygen (O($^1$D) - a product of ozone photolysis). The dominant loss process for CFC-114 is the former,

with the latter thought to be responsible for 25% of its total stratospheric loss. The reaction of CFC-114a with O($^1$D) is thought to be faster than that of CFC-114. In addition, CFC-114a is more susceptible to photolysis than CFC-114, resulting in the CFC-114a atmospheric lifetime being shorter than that of CFC-114 (Burkholder et al. 2013; Davis et al., 2016). The total atmospheric steady-state lifetimes of CFC-114 and -114a are currently estimated to be ~189 and ~100 years respectively (Burkholder et al. 2013; Carpenter and Reimann et al., 2014; Davis et al., 2016).

CFC-114 and CFC-114a are difficult to separate as their boiling points are almost identical. The similarity of their mass spectra complicates even their separate detection with mass spectrometric techniques. Therefore, their abundance is usually reported as a sum of both isomers, assuming a fraction of ~10% of CFC-114a (Carpenter and Reimann et al., 2014).

Two previous studies reconstructed historical trends for CFC-114 or the sum of CFC-114 and CFC-114a using firn air data (Sturrock et al., 2002, Martinerie et al., 2009). Sturrock et al. (2002) used inverse firn modelling techniques constrained with

firn air data from an Antarctic site (Law Dome) and air archive data from an observatory at Cape Grim, Australia (40.7˚S, 144.7˚E; Oram, 1999) to reconstruct a CFC-114+CFC-114a atmospheric trend, and concluded that Southern Hemispheric concentrations were negligible before 1960. Their data were compared with University of East Anglia (UEA) data of CFC-114 (fully separated from CFC-114a) from Cape Grim on an earlier UEA calibration scale (Lee, 1994) and a calibration difference (factor of 0.94, constant over time) was found. Martinerie et al. (2009) used AFEAS emissions and an

atmospheric chemistry model to calculate atmospheric trends that were compared to firn data at 5 sites from Antarctica and Greenland using a forward firn modelling approach. They concluded that the AFEAS emissions based trend, leading to significant atmospheric concentrations before 1960, is inconsistent with the firn and atmospheric data based trend from Sturrock et al. (2002) and that the Sturrock et al. (2002) trend is more consistent with their Northern Hemisphere firn data than the AFEAS based trend. The firn data used in Martinerie et al. (2009) are a combination of UEA CFC-114

measurements at North GRIP, Berkner Island and Dome C (earlier calibration scale) and NCAR CFC-114+CFC-114a measurements at Devon Island, North GRIP and Dronning Maud Land.

In addition to early attempts of quantification (e.g. Chen et al., 1994) and the regular global mixing ratio updates in recent WMO Ozone Assessments, Reimann et al., 2004 reported atmospheric CFC-114 abundances having stabilised with elevated levels essentially absent in the latest part of the record from the high-altitude station at Jungfraujoch, Switzerland. Moreover,

the study of Chan et al., 2007 found no substantial emissions from the heavily industrialised region in the Pearl River delta in China. Again, none of these studies distinguishes between the two isomeric forms of CFC-114. We here provide, for the first time, a complete quantification of both isomers, based on an analysis of a combination of archived remote Southern Hemispheric tropospheric air and firn air data that allows the reconstruction of a tropospheric record from 1960 to 2014. The abundances, temporal evolution and emissions of both isomers are evaluated using measurement and modelling techniques





that have been updated and improved since the first measurements in the 1990s (Lee, 1994 and Oram 1999). Further insights are derived from aircraft-based observations as well as samples collected in East Asia.

## 3 Methodology

### 3.1 Sample collection

Air samples are routinely collected at Cape Grim for measurement and archiving during baseline conditions (clean marine air, wind speeds >15 km h$^{-1}$ and direction 190-290° (Fraser et al. 1999) representing unpolluted Southern Hemispheric air with long air mass back-trajectories over the Southern Ocean, distant from pollution sources prior to their arrival at Cape Grim. UEA stores an archive of such air samples currently spanning 1978-2014. We have analysed 117 of these archived samples. All samples collected before 1994 were sub-samples of the parent Cape Grim air archive, transferred to and stored

in 3 litre electropolished stainless steel canisters. Post 1994 the majority of samples have been collected *in situ* into 3 litre stainless steel canisters (either electrochemically passivated or Silco-treated (Restek Corp.), e.g. Sturges, et al. 2012) to ~3 bar, using a metal bellows pump, with the remainder being sub-samples of the parent archive. Several studies have shown that mixing ratios of many species within the parent CSIRO Archive or UEA sub-archive are similar to in situ measurements made at Cape Grim at or close to the time of the archive sampling; therefore verifying this archive as representative of actual

background air and allowing storage of these samples without substantial alterations of concentrations within the samples (e.g. Fraser et al. 1986; Vollmer et al., 2011; Oram et al., 2012; Laube et al., 2013).

In addition, 39 air samples extracted from deep firn snow during two Antarctic drilling campaigns (Berkner Island & Dome C) enabled the reconstruction of atmospheric histories of the two gases from dates preceding the start of the Cape Grim record. The firn air extraction procedure and the characteristics of the drilling sites are described in Martinerie et al. (2009).

Also, 15 upper tropospheric samples collected by the CARIBIC Observatory (www.caribic-atmospheric.com) on flights between Germany and South Africa on 10th and 11th February 2015 were analysed to assess current interhemispheric mixing ratio gradients and their consistency with the inferred tropospheric records. For details on the sampling system please refer to Brenninkmeijer et al. (2007). We also include results from 23 air samples collected in the important East Asian source region during a ground-based campaign in Taiwan from the Hengchun site (22.1°N, 120.7°E, 7 m a.s.l.) in March and

April 2015.

### 3.2 Analytical technique

Samples were analysed for CFC-114 and CFC-114a by cryogenic trapping followed by Gas Chromatographic separation and Mass Spectrometric detection (GC-MS). The GC-MS method is very similar to that described in detail in Laube et al. 2013. Briefly, samples were dried by passing through a magnesium perchlorate ($Mg(ClO_4)_2$) drying tube. Condensable trace gases

were subsequently trapped in a packed stainless steel sample loop submerged in a cold bath held at -78°C. The sample loop



was filled with Hayesep D adsorbent giving quantitative retention and release. The sample loop was heated to near 100 °C to ensure immediate and complete desorption of the analytes. Some of the samples were analysed on an older but very similar GC-MS system (Oram, 1999; Fraser et al., 1999; Oram et al. 2012; further details in section 3.3).

Separation was carried out using an Agilent 6890 Gas Chromatograph. For full separation of the CFC isomers, a Porous
Layer Open Tubular (PLOT) GC column was used, with aluminium oxide ($Al_2O_3$) as the stationary phase, deactivated by potassium chloride (KCl). This deviates significantly from the method described in Laube et al. 2013 and is the crucial detail of the methodology within this study, as this type of GC column does separate CFC-114a from CFC-114. The two isomers are primarily separated as a function of their polarities, rather than their boiling points. The GC column was held at -10 °C for 2 minutes and then heated to 180 °C at 10 °C/min while maintaining a flow rate of 2 ml/min.

The GC is connected to a high sensitivity trisector (EBE) mass spectrometer (MS) (Micromass/Waters Autospec), which has a typical detection limit < 0.1 femtomole per mole of air ($10^{-16}$) when extracting from 300 ml of air, and was operated in electron impact selected ion recording (EI-SIR) mode, and at a mass resolution of ~1000 at 5% peak height. CFC-114 and CFC-114a were measured using mass fragments $C_2F_4{}^{35}Cl^+$ and $C_2F_4{}^{37}Cl^+$ ($m/z$ 134.96 and 136.96). The retention times were 16.69 and 16.87 minutes for CFC-114 and CFC-114a respectively. A pure, research-grade helium sample ("blank") was
measured on each day and no system contamination was observed of relevance to the analysis of the two compounds. During analysis all samples were bracketed by a "working standard" (clean northern hemispheric air, collected in 2006) after every two to three samples. Measurement uncertainties were calculated as the square root of the sum of the squares of the 1 σ standard deviations of sample and standard measurements. The average precision was 1.1 % for both isomers.

The detector response was evaluated with regard to its linearity using the same methodology as in Laube et al., 2014, i.e.
using a static dilution series prepared from a background air sample collected in 2009 at Niwot Ridge near Boulder, USA (containing 15.2 ppt of CFC-114, and 1.03 ppt of CFC-114a, see 3.3 for calibration) with pure nitrogen in stainless steel canisters. The six dilutions were 100, 67, 30, 15, 7 and 0 % and we found linearity within 1.9 % which is well within the uncertainties of the dilution factors and measurement uncertainties (less than 5 % in all cases).

### 3.3 Calibration

Calibration scales were established for CFC-114 and CFC-114a by a two-step dilution process described in Laube et al., 2012. A pure sample of a mixture of both isomers (5.7 % CFC-114a) was provided by DuPont. This isomeric ratio was determined by Gas Chromatography with Flame Ionization Detection (GC-FID) at DuPont. The calibration sample was diluted into 99.7 litre aluminium drums to near-atmospheric levels (CFC-114: 120 to 160 parts per trillion or ppt; CFC-114a: 6 to 9 ppt) in oxygen-free nitrogen. We here report one particular improvement as compared to the previously reported
calibration system, i.e. the improved leak rate of these drums which was achieved through extensive leak-testing and the use of epoxy resin. Observed internal levels of outside air have been reduced to below 0.01 % thus rendering previously required corrections unnecessary. The dilutions were analysed by GC-MS (described above and in Laube et al. 2012), and used to assign mixing ratios to the above-mentioned internal reference standard provided by NOAA (used as the working standard).





The same dilutions were also analysed in full scan mode to ensure their purity. A CFC of known atmospheric abundance (CFC-12: diluted to between 260 and 290 ppt) was added to the dilution drums to assess accuracy of the calibrations by comparing calculated mixing ratios to NOAA calibration values. The three separate calibration analyses were accurate to within 2.4% of NOAA values (using CFC-12 mixing ratios, 2006 NOAA scale) and had a standard deviation of 1.2 % (CFC-

114) and 1.5% (CFC-114a) respectively. Determined mixing ratios are expressed on a volumetric dilution scale, which is not equivalent to a mass based scale (as used by e.g. NOAA) unless ideal gas behaviour is assumed. The resultant calibration error in assuming equivalence to a molar (mass) scale has however been proven to be negligible for this particular calibration system (Laube et al., 2010).

These new calibrations were also applied to existing data from firn air (only CFC-114 published in Martinerie et al., 2009) as

well as the earlier part of the Cape Grim record (Oram, 1999). Both of these data sets were analysed on a previous version of the GC-MS system with the same type of GC column (which has long been known to separate the two isomers) and also using different air standards. Older data had to be transferred to the new calibration scales using repeatedly measured ratios between internal standards. The conversion factor from the old calibration scale (Lee, 1994) as published in Martinerie et al., (2009) was determined as 0.9185 for CFC-114 (CFC-114a: 0.5808). To ensure comparability of the data sets, 14 Cape Grim

samples collected between 1978 and 2004 have been analysed on both systems and these data agree within uncertainties for both isomers and show no indication for any systematic offset. We therefore conclude that all presented data sets are comparable and can be combined.

### 3.4 Firn modelling

Forward models of gas transport in firn (e.g. Buizert et al., 2012) use an atmospheric mixing ratio trend as input and predict a

concentration profile versus depth in firn, which results from gas transport processes in firn such as molecular diffusion, gravitational setting, wind-driven convection etc. Here we use an improved version of the firn model used in Martinerie et al. (2009). A major upgrade is the use of a firn diffusivity profile which optimally fits data from several reference gases with well-known atmospheric histories in the firn (Witrant et al., 2012). This model performed well in an international inter-comparison study (Buizert et al., 2012). Two species-dependent physical constants are used in the model: molecular mass

and diffusion coefficient in air. We used measured values of the CFC-114 diffusion coefficient from Matsunaga et al. (1993). To our knowledge, no measurement is available for CFC-114a but the estimation methods commonly used in firn models (Fuller et al., 1966, Chen and Othmer, 1962, Marsh et al., 2007) provide the same diffusion coefficient for CFC-114 and CFC-114a within uncertainties, which we therefore use here.

Inverse models of gas transport in firn use mixing ratio measurements in firn as input and predict atmospheric trends. Such

an inverse approach was applied to CFC-114 by Sturrock et al. (2002). Here we use a recently improved inverse model (Rommelaere et al., 1997; Witrant and Martinerie, 2013) which can be constrained by several firn air sampling sites at the same time. The firn model improvements combined with the optimal inverse fit of the data lead to a much better agreement (Figure S1) between the calculated atmospheric trend and firn data than in Martinerie et al. (2009). On the other hand, it does



not allow an evaluation of the consistency of firn data with emission based trends. In order to discuss the CFC-114 and CFC-114a budgets, we use an inverse (or top-down) atmospheric modelling approach to infer emissions from atmospheric concentrations (see next section) rather than the forward atmospheric modelling approach in Martinerie et al. (2009).

### 3.5 Emission modelling

The top-down global annual emissions estimates of the CFC-114 and CFC-114a were derived using a two-dimensional atmospheric chemistry-transport model. The model comprises of grid boxes which have been equally divided into 24 equal-area, zonally-averaged bands and has 12 vertical layers of 2 km depth. The latitudinal distribution of emissions is based on the assumption that 95% of emissions originate from industrial activities in the Northern Hemisphere, predominantly from mid-latitudes. By using these preferred latitudinal distributions, the transport scheme of the model has been shown to
reproduce the reported global distributions of CFC-11 and CFC-12 within 5 % (Reeves et al., 2005).

For the photolysis of CFC-114, the absorption cross sections are calculated for each grid box as a function of seasonally varying temperature for the wavelengths 200 – 220 nm (Simon et al. 1988). A log-linear extrapolation of the Simon et al. (1988) data, $\log \sigma (\lambda) = -1.8233-0.0913\lambda$ was used to derive the absorption cross sections for longer wavelengths in the range of 222 – 235 nm (Sander et al. 2011). Due to the unavailability of UV absorption spectrum data reported for CFC-114a at the
time, the same absorption data were used for CFC-114 and CFC-114a. The rate coefficients of $1.43 \times 10^{-10}$ $cm^3$ molecule$^{-1}$ s$^{-1}$ and $1.62 \times 10^{-10}$ $cm^3$ molecule$^{-1}$ s$^{-1}$ are applied for the reaction of O($^1$D) with CFC-114 (Baasandorj et al., 2013) and CFC-114a (Baasandorj et al., 2011), respectively. The recommended values mentioned above are based on work by and. The diffusive loss from the top of the model was set by adjusting the mixing ratio of our studied compounds at 25 km (i.e. the boundary conditions) such that they were a fraction (F) of those in the top model box (23 km) (Newland et al. 2013). We use
values of 0.922 and 0.826 for F in order to achieve the steady state lifetime of 189 and 100 years for CFC-114 and CFC-114a, respectively, based on the estimates recently been reported in SPARC (2013) and Carpenter and Reimann et al., (2014). This is in agreement with the very recently reported lifetime of 105.3 years for CFC-114a, which took into account new UV absorption data (Davis et al., 2016). It should be noted that the vast majority of the loss of both CFC-114 and CFC-114a occurs above the height of the model domain so their modelled lifetimes are largely controlled by the values assigned
to F, which is adjusted to give the reported atmospheric lifetimes. Once the model had been set up with the above conditions, the emissions in the model were iteratively adjusted (Newman et al., 2013) until the predicted concentrations matched the Cape Grim air measurements from 1978 to 2014. The same approach was also applied using the Southern Hemisphere firn air-derived trends from 1960 to 2003.

The determination of uncertainty ranges of the emission estimates of CFC-114 and CFC-114a were based on the
combination the average measurement uncertainty (1.1%), the modelling uncertainty (5%) and the model fit uncertainty (1.1%) (combined as the square root of the sum of squares of individual uncertainties). Then the calibration uncertainty (2.4 %) was added to this to give an overall uncertainty. The calculated uncertainty ranges (± 7.6%) were added to the 'best fit' modelled mixing ratios for Cape Grim to derive an envelope of upper and lower uncertainty bounds. The model was then





rerun to fit to the upper bound of this envelope using the lower estimate of the lifetime to give the maximum emissions and similarly rerun to fit to the lower bound of the envelope using the upper estimate of the lifetime to give the minimum emissions (following the methodology of Kloss et al. 2014). The range of lifetimes used for CFC-114 was 153-247 years for CFC-114 (SPARC, 2013; Carpenter and Reimann et al., 2014) and a similar relative range of lifetimes was assumed for

CFC-114a.

## 4 Results and discussion

### 4.1 Tropospheric long-term trends from firn air and the Cape Grim archive

The temporal evolution of CFC-114 and CFC-114a is shown in Figure 1. The Southern Hemispheric trend reconstructed from firn air reveals that atmospheric abundances of both isomers became significant in the 1960s with accelerating

abundances until the late 1970s. The CFC-114 record is similar to that presented in Sturrock et al., 2002 (Figure S1), who also used firn air reconstructions from a different Antarctic site at Law Dome, and an "early day" inverse modelling technique. From both air archives (firn and Cape Grim) we find a further steady increase in abundance from 1978 until the 1990s, followed by a weakening in growth. Also apparent from Figure 1 is that mixing ratios of CFC-114 stopped increasing around 1993 while those of CFC-114a continued to increase until around 2000. This will be discussed further in section 4.2.

The average atmospheric abundances of CFC-114 and CFC-114a at Cape Grim in 2012 were 15.2 ± 0.3 ppt and 1.05 ± 0.01 ppt respectively. This means that our result agrees well with the combined mixing ratio of 16.33 ppt given in Carpenter and Reimann et al. (2014) at this point in time (i.e. 2012), with the corresponding calibration scale having been first reported in Prinn et al. (2000). However, our Cape Grim record reveals a steadily increasing contribution from CFC-114a starting at 4.2 % in 1978 (Figure 2) and reaching 6.9 % in 2014. This is confirmed by the ratios observed in the firn air-derived record,

which shows pre-1978 CFC-114a/CFC-114 ratios of well below 4 %, although with considerable uncertainties (Figure 2). Therefore the ~10% contribution of CFC-114a that has been assumed in Carpenter and Reimann et al. (2014) and previous WMO/UNEP Ozone assessments appears to have been an overestimate. Moreover, the contribution of CFC-114a to the sum of the isomers has been assumed constant in those previous assessments, which is clearly not the case.

CFC-114 is, at the end of our record in 2014, the fourth most abundant CFC in the atmosphere (after CFC-11, CFC-12 and

CFC-113). Its mixing ratios were, on average, decreasing at 0.01 ppt/year between 2008 and 2014. This is in agreement with Carpenter and Reimann et al. (2014) who reported an average decrease of 0.01 ppt/year between 2008 and 2012. Although being the fourth most abundant CFC, CFC-114 mixing ratios are substantially lower than the three others (CFC-11: 236.3, CFC-12: 524.4 and CFC-113: 73.8 ppt, NOAA global average in 2012, Carpenter and Reimann et al., 2014). Its isomer CFC-114a is found to be the seventh-most abundant CFC in the atmosphere, after CFC-115 and CFC-13, with its growth rate

not turning negative until 2008 with a subsequent average decrease of 0.001 ppt/year.





### 4.2 Global emission estimates

The atmospheric abundances of CFC-114 and CFC-114a are indicative of their accumulated emissions into the atmosphere. Figure 3 shows the model based reconstructed global annual emissions of both isomers as derived from the Cape Grim observations. Emissions were already high for both isomers at the beginning of the record in 1978 and peaked between 1986 and 1988 at 18.5 Gg/year (CFC-114) and 1.7 Gg/year (CFC-114a).

Figure 3 shows very similar emission behaviour of both isomers until around 1991. This similarity in the time series of annual emissions is consistent with the use of the isomers as a mixture leading to co-emission. This is even more apparent when looking at the ratio of their emissions (Figure 2) which remained nearly constant at around 9 % between 1978 and 1991. Such a constant emission ratio may seem counterintuitive at first as the observed ratio of the mixing ratios of the two isomers increases rapidly throughout that period (also shown in Figure 2). In addition CFC-114a (100 years, Carpenter and Reimann et al., 2014) has a much shorter atmospheric lifetime as compared to CFC-114 (189 years, SPARC, 2013). These two facts imply that increasingly higher emissions of CFC-114a would be needed to sustain increases in mixing ratios above those of CFC-114. Both effects (i.e. the increasing ratio of mixing ratios and the lifetime difference) are however compensated in the emissions by the fact that the emission ratio of CFC-114a/CFC-114 between 1978 and 1991 is at about 9 % well above the ratio of the mixing ratios over the same period, which rises from 4.2 to 6.5 %. The current assumption of isomeric mixtures emitted to the atmosphere containing ~10 % of CFC-114a is consistent with this 13-year period. However, the implication is that the ratio of pre-1978 emissions must have been significantly more biased towards CFC-114. In other words, emission prior to 1978 must have largely consisted of more than 96% of CFC-114 and 4 % or less of CFC-114a. This is confirmed by ratio of the mixing ratios in both the Cape Grim-based and the firn-based records (Figure 2) and could point to a change in manufacturing processes or partly independent source(s).

From 1991 onwards we find a sharp increase of CFC-114a emissions relative to those of CFC-114. While emissions of both isomers decrease substantially throughout the 1990s, those of CFC-114a decline much slower. The isomeric emission ratio (Figure 2) only starts to decrease again after CFC-114 emissions stop declining in 1996. In contrast to CFC-114, emissions of CFC-114a continue to decline until 2010. This may seem surprising at first, but could perhaps be reconciled by the aforementioned involvement of pure CFC-114a in the production of HFC-134a (Banks et al. 1994). Incidentally, abundances of HFC-134a started increasing in the atmosphere in the early 1990s (Montzka et al., 1996b; Oram et al., 1996) as it replaced CFCs predominantly in mobile air conditioning. However, our CFC-114a emission data do not suggest that it is an impurity in all the HFC-134a produced as mixing ratios of the latter continue to increase to date (Carpenter and Reimann, 2014). CFC-114a is only an intermediate in one of the pathways to synthesise HFC-134a. Our CFC-114a emission data are consistent with two possible scenarios, i.e. a) emissions of CFC-114a as an impurity in HFC-134a produced via that pathway, as well as b) emissions at the HFC-134a production level.

We also compare the emission estimates from our "top-down" observation-based approach with the "bottom-up" emissions derived from production and release data from the Alternative Fluorocarbons Environmental Acceptability Study (AFEAS)





in Figure 4. We are only able to compare the sum of both isomers as AFEAS does not distinguish between CFC-114 and CFC-114a. The bottom-up data start much earlier in 1934 but also end earlier in 2003. This is due to reporting companies responding to AFEAS that from 2003 onwards CFC-114 represented a small and diminishing fraction of global CFC production, which resulted in no further CFC data being sought or reported (AFEAS, 2009).

In order to be able to compare bottom-up and top-down emissions prior to 1978 an extra emission model run was carried out, matching the firn-derived pre-1978 trend to the Cape Grim-derived record in 1978, which was successful within the constraint from the mixing ratio uncertainty ranges of the firn air record (Figure 4). The exact temporal shape of this pre-1978 emission record is very uncertain as the uncertainty range in the firn air-derived mixing ratios (Figure 1) allows a large range of growth rates and therefore emission scenarios. Nevertheless it enables us to conclude that the early part of the

AFEAS data, which suggests rapidly increasing emissions to more than 5 Gg/year in the late 1940s, are inconsistent with our emissions. These emissions are very unlikely to have occurred before the mid-1950s. Moreover, a pre-1978 emission maximum is required for CFC-114 (but not for CFC-114a) in order to satisfy both constraints i.e. a) matching the emissions of the Cape Grim and firn records in 1978 and b) not leaving the firn-based uncertainty range. When comparing with existing literature it is notable that the early AFEAS record agrees with the mixing ratio time series of CFC-114 published in

Martinerie et al. (2009). This is however mostly because that study used AFEAS emissions as a prior input to the inversion (see also section 3.5); while our records are very similar to the top-down approach-based data set published in Sturrock et al. (2002).

In the overlap period of AFEAS with our Cape Grim-based post-1978 record we find agreement between the two data sets within our uncertainty range apart from a period in the early 1990s. From 1990 to 1993 our emissions are significantly

higher than the AFEAS data (Figure 4). It should however be noted that, while no uncertainties are given in the AFEAS data base, there are considerable uncertainties related to bottom-up methods, which are difficult to quantify. This especially applies to the timing of the release to the atmosphere. Differences between the two emission data sets (release data for AFEAS) reach up to 5.0 Gg/year in 1991 but this discrepancy all but disappears after 1993. Both data sets also agree that emissions decreased rapidly and stabilised between 2.1 and 2.3 Gg/year from 2000 onwards, demonstrating the success of

the Montreal Protocol. Cumulative emissions from our top-down approach reach 530 Gg in 2003 (uncertainty range from 433 to 631 Gg) and agree very well with both AFEAS production and release figures between 1934 and 2003, which have been reported at 520 Gg and 511 Gg respectively (AFEAS, 2009). The aforementioned discrepancy in the early part of the record may therefore well originate from pre-1960 production reported by AFEAS which was not immediately released to the atmosphere. For the post-2003 part, as discussed above and according to AFEAS, a substantial amount of CFC-114

(containing a fraction of CFC-114a) has been believed to be in banks of long-lived equipment. The AFEAS data base itself does however not fully reflect this in their emissions as only 8.8 Gg remain "unreleased" to the atmosphere in 2003. If current emissions are from existing equipment, then such a small 'bank' is not consistent with current persisting emissions of 1.80 Gg/year (range: 1.0 to 2.7 Gg/year) of CFC-114 and 0.25 Gg/year (range: 0.16 to 0.35 Gg/year) of CFC-114a in 2014,




i.e. 11 years after a remainder of 8.8 Gg of banks has been reported. In contrast our cumulative emissions for those 11 years amount to 23 Gg (13 to 34 Gg).

Finally we have also derived emissions purely based on the atmospheric records derived from the firn air data (dotted lines in Figures 3 and 4). This illustrates the limitations of that methodology when relying on data from only two sites. Atmospheric mixing ratio records fit those from Cape Grim well for the overlap period (Figure 1). Emission estimates do however strongly depend on the annual growth rates, and thus small discrepancies between the curvature of the firn-derived and the Cape Grim-based atmospheric records translate into large changes in the temporal distribution of the estimated emissions. The limited accumulation rate of the two firn sites used here prevents a high temporal resolution of the respective record and results partly in smoothing and partly in a shift of the timing of the emissions. However, the total emissions estimated from the firn record are 530 Gg (range: 505 to 557 Gg, period from 1960 to 2003) which agrees very well with those from the mostly Cape Grim-based trend over the same period, as well as the AFEAS data when including emissions reported from 1934 onwards.

### 4.3 An interhemispheric gradient case study from CARIBIC

Most industrialised countries are located in the Northern Hemisphere, which is why trace gases of predominantly anthropogenic origin are known to show interhemispheric gradients (e.g. Carpenter and Reimann et al., 2014). The results from interhemispheric flights by the CARIBIC Observatory are shown in Figure 5. Even though we find slightly higher mixing ratios in the Northern Hemisphere, the gradient with latitude is neither significant for CFC-114 nor for CFC-114a (within the 1 σ measurement uncertainty, i.e. 1.2 % on average for both gases; compared to gradients of 0.8 % and 1.0 % for CFC-114 and -114a respectively when looking at the variability of the atmospheric mixing ratios averaged over both flights). This is consistent with the Cape Grim data that show that global emissions of both of these gases have largely ceased. As our GC-MS analyses revealed no exceptionally high mixing ratios of many other trace gases (e.g. CFC-11, H-1301, HCFC-142b, HFC-134a), we conclude that the sampled air masses are representative of well mixed mid and upper tropospheric background air during February 2015.

### 4.4 Tropospheric samples from Taiwan

Figure 6 shows the results of our measurements of both isomers on samples collected during a field campaign in southern Taiwan in 2015 (similar to the 2013 campaign reported by Vollmer et al., 2015). The observed air masses mostly reached the sampling site from China and the Korean Peninsula with no significant influence from Taiwanese industrial regions. Interestingly, we find up to 17% higher mixing ratios of CFC-114a when comparing with average mixing ratios observed at Cape Grim between 2012 and 2014. Even samples that show no significantly elevated mixing ratios for several other trace gases that are known to have continuing strong East Asian sources (e.g. HCFC-141b, HFC-227ea) exhibited CFC-114a mixing ratio more than 2% higher than at Cape Grim. In contrast mixing ratios of CFC-114 are not enhanced significantly throughout the campaign confirming that the regional source of CFC-114a is not due to the emission of an isomeric mixture.





HFC-134a during this campaign showed mixing ratios close to background (6 samples between 84 and 88 ppt) as well as enhancements of up to 132 ppt. However, when comparing CFC-114a and HFC-134a we find no significant correlation ($R^2$ < 0.1), implying that either i) most of the regional HFC-134a emissions originate from production pathways not involving CFC-114a and/or ii) HFC-134a does not contain CFC-114a as an impurity and the latter is only emitted during HFC-134a

production. The connection of these regional CFC-114a emissions to HFC-134a production processes is however supported by the fact that we see the biggest enhancements of CFC-115 (between 5 and 10 % above background) and CFC-113a (between 90 and 200 %) in the 4 samples with the highest CFC-114a mixing ratios; with both these compounds being involved in the same HFC-134a production process (where CFC-113 is isomerised to form CFC-113a, which is then fluorinated to produce CFC-114a, followed by hydrogenolysis to HFC-134a, CFC-115 being a small by-product as a result

of overfluorination, Banks et al., 1994). In addition we cannot rule out the possibility of a new onset of CFC-114a emissions as the Taiwan samples were collected after the end of our current Cape Grim record.

## 5 Conclusions

These tropospheric observations provide, for the first time, long-term trends and emissions of both CFC-114 and CFC-114a. Based on firn air reconstructions from two Antarctic sites, both isomers had very low atmospheric abundances (< 0.3 ppt)

before 1960, which is in agreement with an earlier study that reported CFC-114 with an undetermined fraction of CFC-114a and was based on a different firn drilling site (Sturrock et al., 2002). We demonstrate the impact of the Montreal Protocol regulations, which banned consumption in developed countries from 1996 (UNEP, 2014) and this is probably the driver of the stabilisation of the global atmospheric mixing ratios of both CFCs. We estimate global cumulative emissions of 553 Gg (range: 415 to 617 Gg) of CFC-114 and 39.5 Gg (31 to 48 Gg) of CFC-114a up until 2014, which is largely consistent with

industry estimates (AFEAS, 2009); except for the timing of the early emissions and the remainder in banks after 2003. When using the published Global Warming Potentials (GWPs) of CFC-114 (GWP-100: 8490, Carpenter and Reimann et al., 2014) and CFC-114a (GWP-100: 6510, Davis et al., 2016), we calculate that these cumulative emissions are equivalent to the emission of 4.67 billion tonnes of $CO_2$. As a result of its higher GWP and its higher abundance CFC-114 is the dominant contributor (94.5 %) to these $CO_2$-equivalent emissions.

We also find that significant global atmospheric emissions of 1.8 Gg/year (CFC-114, range: 1.0 to 2.7 Gg/year) and 0.25 Gg/year (CFC-114a, range: 0.16 to 0.35 Gg/year) persisted until at least 2014, highlighting the need for continued efforts to ensure that these substances eventually disappear from the atmosphere. Further observations are also required to understand the origin of those emissions, especially in the East Asian region. It should however be noted that such emissions are not necessarily in breach of the Montreal Protocol given that CFCs used as intermediates in the production of other compounds

(such as HFC-134a) do not have to be reported under that treaty.

Importantly, CFC-114 and CFC-114a were not always co-produced or co-emitted. This results in time-dependent changes in the ratio of the isomers in atmospheric samples. Thus the use of a simple time-invariant correction (in %) as assumed in



recent assessments of climate change and ozone depletion (e.g. Myhre et al., 2013, Carpenter and Reimann et al., 2014) is not correct when discussing their abundance changes over time and their impacts on ozone depletion and radiative forcing. Finally, given the differences in trends and emissions we recommend that the two isomers should be reported separately in the future, or that time-dependent speciation factors, such as presented here, should be used to approximate global

concentrations of CFC-114 and CFC-114a.

*Acknowledgements.*

J.C.L. received funding from the UK Natural Environment Research Council (Research Fellowship NE/I021918/1) and D.E.O. from the National Centre for Atmospheric Science. The emission modelling work was conducted by N.M.H. through

a PhD studentship funded by the Ministry of Education Malaysia (MOE) and Universiti Kebangsaan Malaysia (UKM). Taiwan-related work was also supported by the NERC IOF award NE/J016012/1 and a NERC Studentship to L.J.G.. We also thank DuPont for providing a CFC-114/CFC-114a mixture, all CARIBIC partners for their contributions and M.J. Newland for constructive discussions. The collection and curation of the Cape Grim Air Archive is jointly funded by CSIRO and the Bureau of Meteorology (BoM); BoM/CGBAPS staff at Cape Grim were/are largely responsible for the collection of

Archive samples and UEA flask air samples; the original (mid-1990s) sub-sampling of the Archive for UEA was funded by AFEAS and CSIRO, ongoing sub-sampling by CSIRO. The Berkner Island drilling was organized and conducted by the British Antarctic Survey with funding from the Natural Environmental Research Council (NERC). Firn air analysis at Dome C contributed to the European Project for Ice Coring in Antarctica (EPICA), a joint ESF/EC scientific programme (ENV4-CT95-0074). The fieldwork at Dome C was also supported by the French Polar Institute (IFRTP) and the ENEA Antarctic

Project (Italy). The firn air work was also funded by the CEC programmes: EUK2-CT2001-00116 (CRYOSTAT) and ENV4-CT97-0406 (FIRETRACC).

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



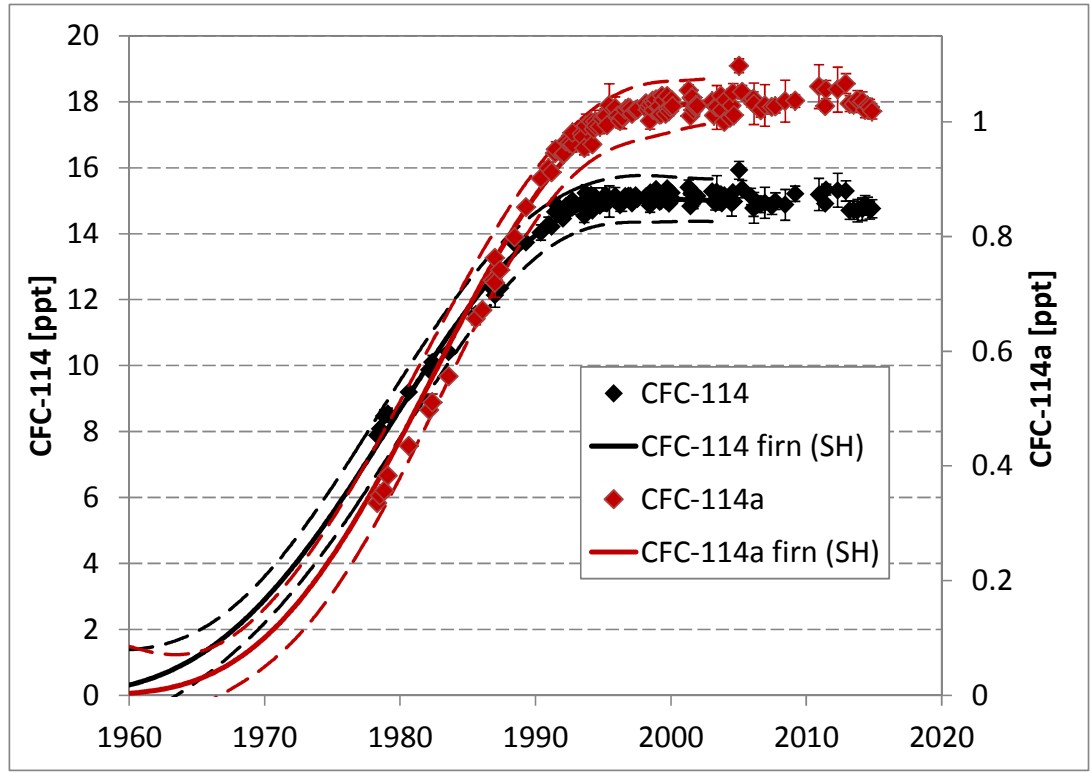

**Figure 1:** Mixing ratios of CFC-114 and CFC-114a as measured in air samples collected at Cape Grim, Australia between 1978 and 2014 (diamonds) and derived from two Antarctic firn air profiles (lines). Uncertainties are 1 σ standard deviations for Cape Grim data and a combination of the former and a firn modelling uncertainty for the latter (shown as dashed lines, see text for further details).



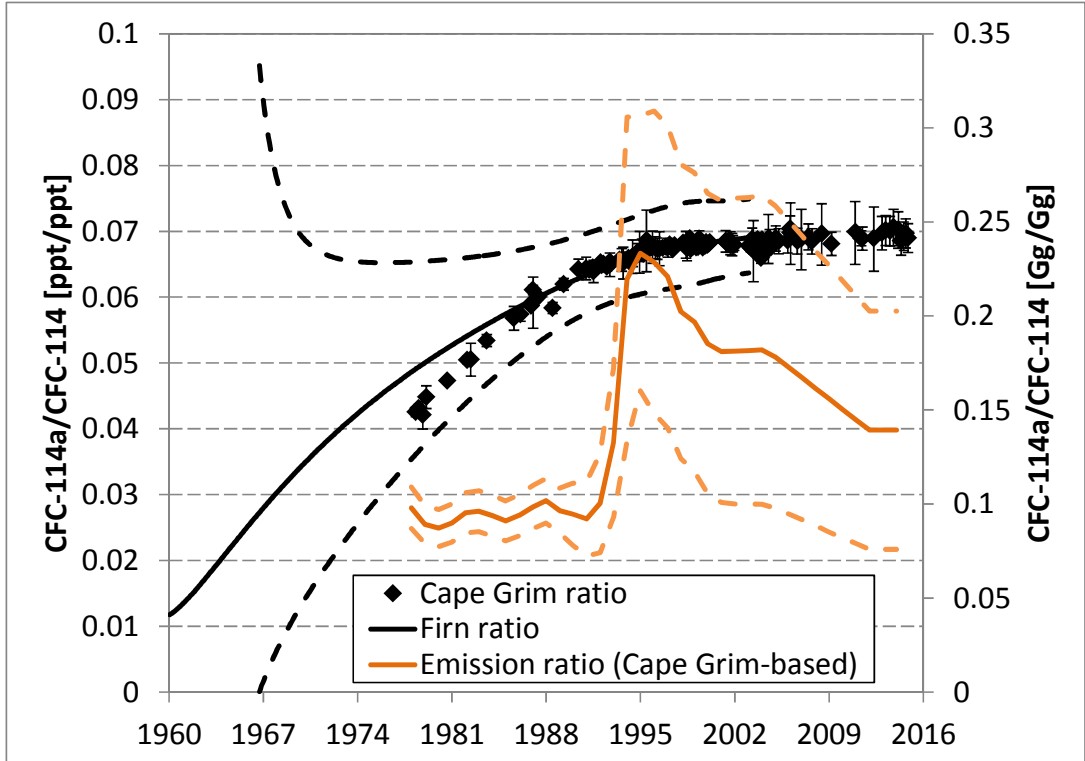

**Figure 2:** CFC-114a/CFC-114 ratio of mixing ratios at Cape Grim (left axis, black diamonds) and derived from Antarctic firn (left axis, black line), as well as the ratio of their emissions derived from these observations (right axis, orange line). Uncertainties of those ratios are calculated from the uncertainties of mixing ratios (Figure 1) and emissions (Figure 3) and
5  using the equation for combining errors in quotients.



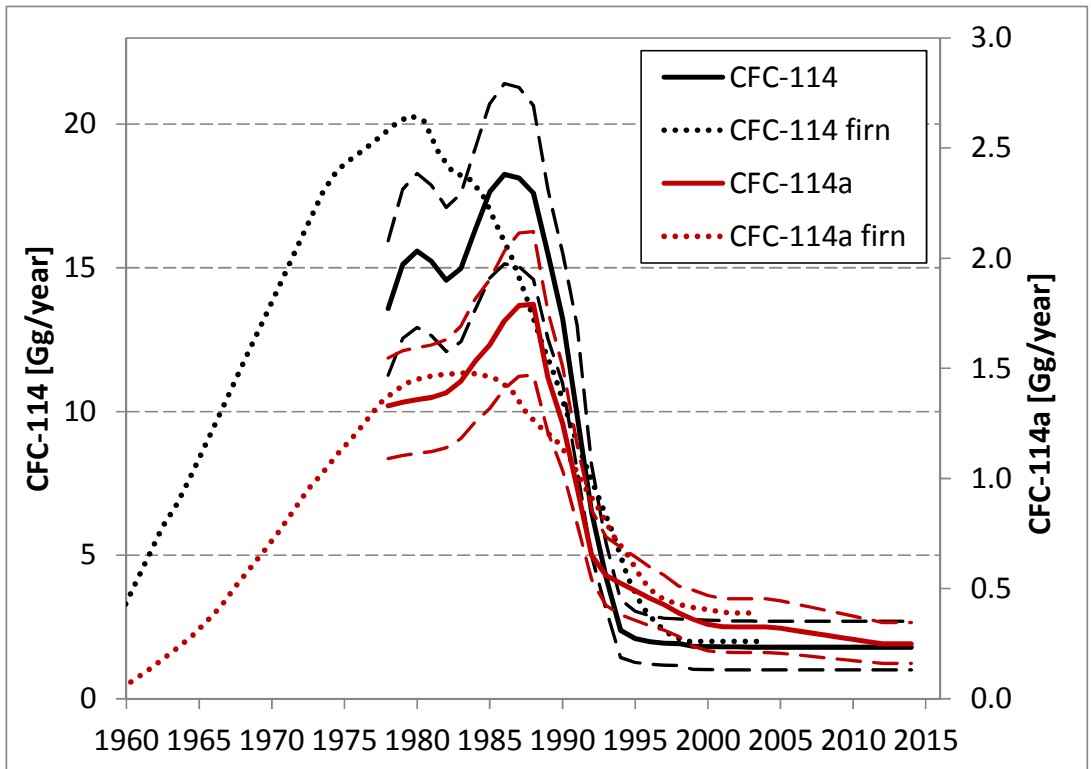

**Figure 3:** Global emissions of CFC-114 and CFC-114a derived from the Cape Grim observations. The dotted black lines are emissions purely derived from firn air observations with dashed lines representing uncertainty ranges.




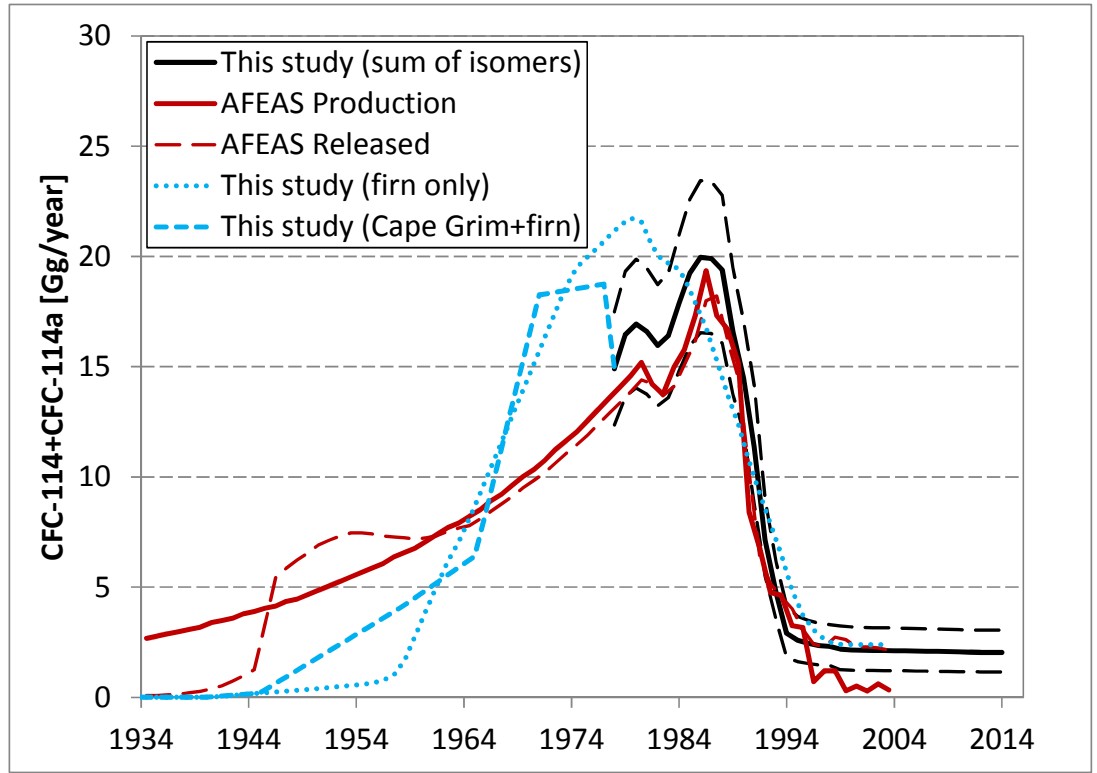

**Figure 4:** Atmospheric observation-based emissions of the sum of CFC-114 and CFC-114a (black line with black dashed lines representing uncertainty ranges) in comparison with "bottom-up" emissions from the AFEAS inventory. The dotted blue line are emissions purely derived from firn air observations while the dashed blue line results from matching the firn-based emissions to the Cape Grim-based emission record in 1978 while not leaving the uncertainty range of the firn-based mixing ratios from Figure 1.





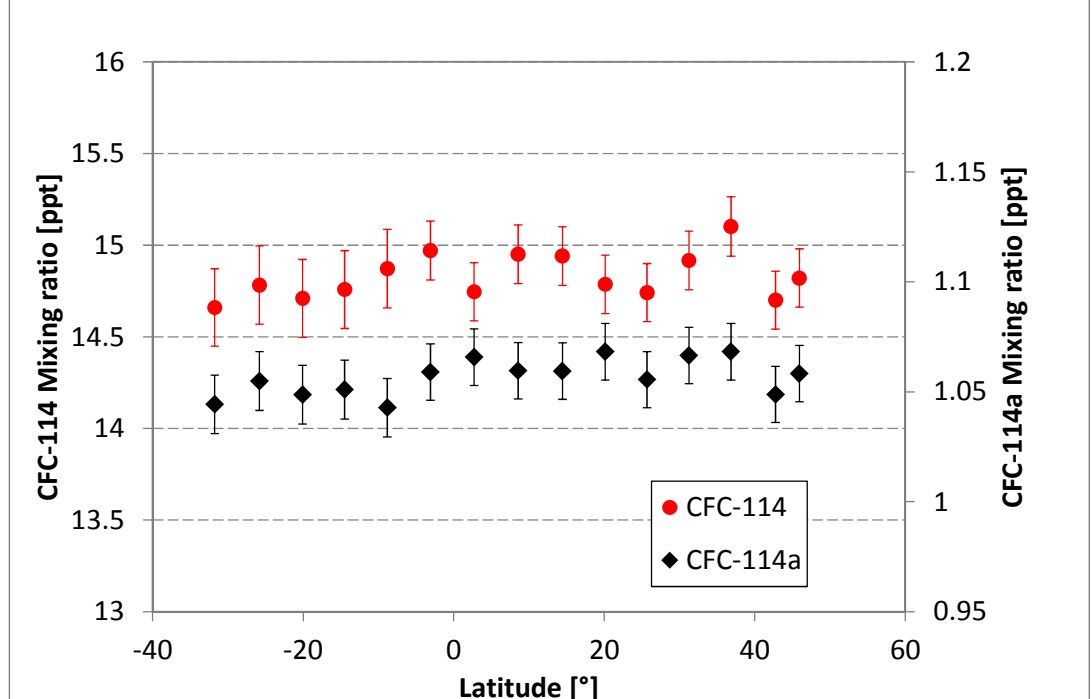

**Figure 5:** CFC-114 and CFC-114a observations from air samples collected during two interhemispheric aircraft flights from Germany to South Africa and back on 10-11th February 2015. Samples were collected between 8.3 and 11.6 km altitude and between 45.9°N (positive values) and 31.8 °S (negative).





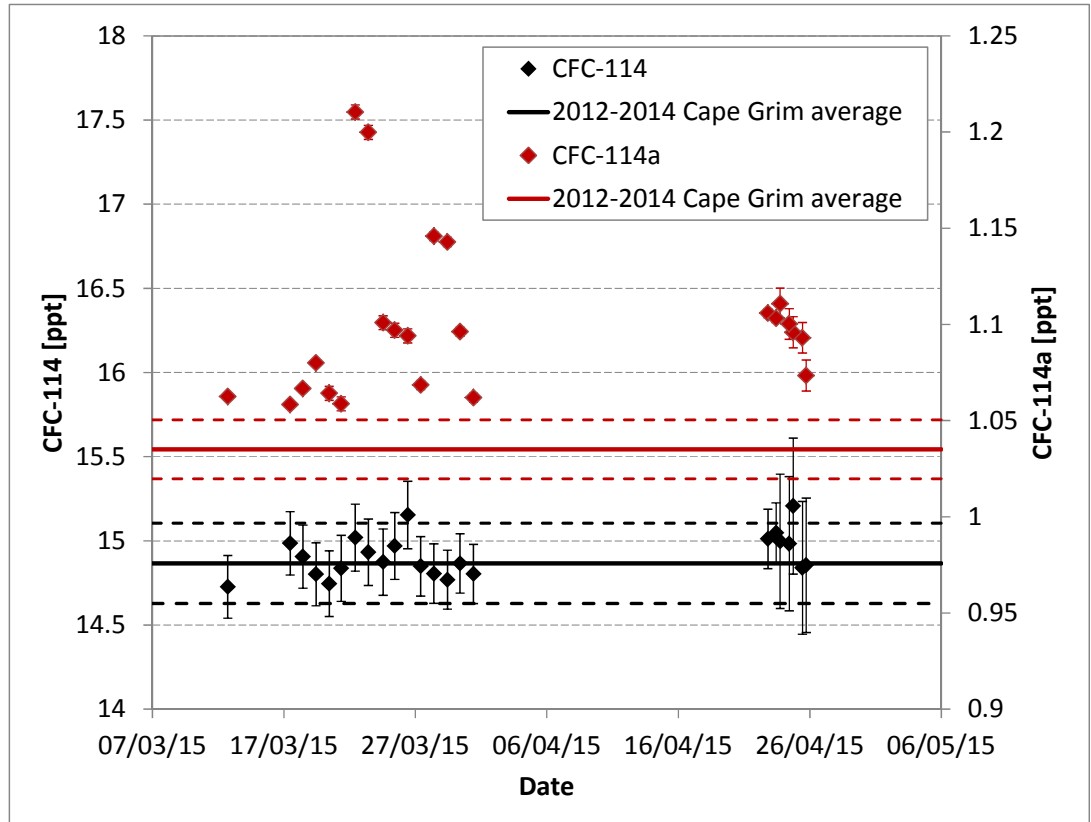

**Figure 6:** Mixing ratios of CFC-114 and CFC-114a from samples collected during a ground-based campaign near Hengchun, Taiwan in early 2015 (diamonds) compared to mixing ratios observed at Cape Grim averaged from 2012 to 2014 (lines). Uncertainties (error bars and dashed lines) are 1σ standard deviations.