# Peer review of "Tropospheric observations of CFC-114 and CFC-114a with a focus on long-term trends and emissions"

_Atmospheric Chemistry and Physics, 2016_

## Referee Comment (RC1) · Anonymous Referee #1 · 1 Sep 2016

General Comments:

This manuscript discusses the determination of isomers CFC-114 and CFC-114a in the atmosphere. The isomers have routinely been reported as a sum of the isomers in the past with the assumption that CFC-114a abundance is $\sim$ 10% that of CFC-114. This manuscript describes how the isomers are separated and how analysis of a variety of atmospheric samples indicates that in fact the source and trends of each isomer have varied over time.

The manuscript provides useful information that is currently missing from many ozone assessment reports.

The presentation of the manuscript and quality of Figures/Tables is appropriate for ACP

Specific Comments:

P1 L22 The global ban came into force in 2010 but non-A5 countries in 1996.

P1 L36 increased from 4.2% to 6.5%

P1 L37 36 year period

P3 L30 Chan et al., 2006 not 2007

P4 L6 missing a closing bracket ) to close (clean marine ...)

P5 L5 Can you provide the details of the column supplier, diameters, length and film thickness.

P4 L16 Fraser at al., 1986 is not listed in the references?

P5 Calibration – what calibration scale are CFC-114 and CFC-114a reported on?

P5 L18 Is it reasonable that the average sample precision is 1.1% for both isomers considering one isomer is 14-22x more abundant than the other, and the mole fraction of the samples has increased from 7.9 to 14.8 (CFC-114) and 0.35 to 1.03 (CFC-1114a) over time.

P5 L26 What is the quoted uncertainty associated with the DuPont sample?

P5 L31 Use of Epoxy resins cover a wide range of potential chemicals and by-products - are the authors sure that no contamination has been introduced to the calibration drums?

P7 L8 On what evidence is the assumption about latitudinal distribution of CFC-114/114a made?

P7 L17 There appears to be some missing text? "based on work by and".

P7 L29-32. The quoted uncertainties appear quite small especially 5% for modelling
uncertainties. Can the author describe more fully how uncertainties have been derived?

P14 L30 need full author list not just et al.:

Figure 1. The way the two y-axes are plotted makes the CFC-114 and CFC-114a trends appear to converge between 1980-1990. Can the scales be adjusted to allow the trends and uncertainties to be viewed more easily?

Figure 2. How can you be sure that the rapid change in ratio of CFC-114a/CFC-114 is not driven by emissions from Asia/Taiwan as detailed in Figure 6?
* * *

---

## Referee Comment (RC2) · Anonymous Referee #2 · 5 Sep 2016

The focus of this paper is the isomeric pair CFC-114 and CFC-114 of these long-lived potent ozone-depletion compounds, which have been regulated by the Montreal Protocol. Samples of firn air and the Cape Grim Air Archive are analyzed for this study, many of which have been analyzed earlier on somewhat different instrumentation. The measurement results are used in combination with a 2-D model to reconstruct atmospheric histories of these compounds, and to assess their global emissions. The data sets are complemented by those from measurement campaigns in Taiwan and from CARIBIC. A focus of the manuscript is put on the relative abundances and emissions of the two isomers, for which the authors find a varying ratio over time.

This study is very well designed and carried out, delivering solid data sets and an distinction into the two isomers. The study is important in several aspects, it allows for a reconstruction of the CFC-114/a histories and emissions including most recent dates and highlights the importance of this so-far rather 'neglected' CFC. The separation into the two isomers reveals some additional new information and presents some new puzzles over the finding of a variable ratio of the emissions and the finding of preferential regional emissions of the CFC-114a over CFC-114. The manuscript is also well written. I have no objections to the scientific interpretation and discussion. I recommend publication after minor revisions.

Comments:

My only major comment is at the very end: publish the numerical results in the Supplements.

page 2, line 28: CFC-114a used in HFC-134a production: Could the authors (here or when discussing the Taiwan results) provide information (perhaps through the later-mentioned contact at DuPont) on the various production pathways for HFC-134a, which is currently the dominant production method, where (geographically) is HFC-134a produced (presumably mainly China?). These comments apply to here or page 9, line 29 (which pathway is dominant?).

page 3, line 13, this paragraph should be revised as it is somewhat unclear and confusing: Shouldn't Oram (1999) be mentioned here as one of the previous studies on CFC-114/CFC-114a — what did Oram (1999) find out related to the two compounds? Not having access to Lee (1994), p. 3, line 18, it is not clear, if Lee (1994) made measurements of these compounds or if that thesis dealt only with some calibration scales for the compounds under discussion.

line 17 ff. The sentence implies that 'the data compared' were Cape Grim measurements, but wouldn't these be the same data in both cases, those published by Oram (1999) except that i.e. Sturrock et al., 2002 used CFC-114 + CFC-114a while those from UEA (also Oram 1999) separated the two compounds. It would be missleading to

call this a 'pure' calibration difference.

page 3, line 28, line 30: Years should be in parenthesis, check entire manuscript for this deficiency.

section 3.2, Analytical technique: could you provide information on potential differences of the molar sensitivities of the two compounds for the two measured mass fragments. Are e.g. for mass 134.96, the peak heights (or areas) per mol of similar size for the two isomers? This information would help to understand potential deficiencies for instrumentation that make combined CFC-114 + CFC114a measurements if reference material differed greatly in composition compared to air samples.

section 3.3 Calibration: pure mixture by DuPont: Could the authors (within confidentiality agreements) provide more details on when (which year) the sample was obtained, if the CFC-114a impurity in CFC-114 production may be constant, which factory it may have been produced (when)?

page 6 line 1: why is 'of known atmospheric abundance' important? Shouldn't it rather say that a compound was used for which independent calibration exists. Was the comparison with NOAA (2.4%) giving a systematic offset for all 3 samples, and which way, which CFC-12 was higher, the calculated or measured? If the 2.4% is considered to be the accuracy for this CFC-114/a calibration scale, it should be stated clearly (the discussion of uncertainty in a later part of the manuscript suggests so).

page 6, line 10: One cannot analyze a 'data set' on a GCMS.

p. 6, line 14: What about the data by Oram (1999), how were these converted? Was Oram (1999) on a preliminary scale? If so, would Lee (1994) mention the difference between scale defined in that thesis and the data published by Oram (1999)? Could the Oram (1999) data be directly converted to the new UEA scale, and if so, what factor, and is it linear. Does UEA plan to give a name to the 'new' scale so to avoid confusion between the past and potential future calibration scales?

page 7, line 14 'at the time'. Why has the model not been re-run with newly-available data? Can the authors assess the error involved with that?

page 7, line 17: incomplete sentence.

page 7, line 24: Can you (perhaps in parenthesis) add the numerical value for the height of the model domain?

page 8, line 3: double mentioning of 'for CFC-114'.

page 8, line 4, 'similar relative range'. Could you be more quantitative?

page 9, line 28: wouldn't it be more appropriate to write 'as emissions of the latter continue to increase'.

page 11, line 26: Provide station name and coordinates.

page 12, line 21: Arent the GWPs given by Harris et al. (vs Carpenter et al)? I couldn't find the value of 8490 in either of the two publications.

references: several obvious errors, check ms carefully: Baasandorj 2013: space missing; Buizert: 55 Sturges; Marsh: lower case words if a journal article, same for other references (e.g. Laube 2014, Sturges (2012); Subscripted numbers in chemical formulae (Oram 1999), Oram (2012)

Figures: The figures would greatly benefit from red-coloring of the axis label numbers and text for CFC-114a

Supplement: Provide a title for the Supplement including the manuscript title, authors etc.

Supplement: Provide numerical results of all flask measurements and the major calculation results (e.g. yearly emissions and uncertainty bands).

---

## Referee Comment (RC3) · Anonymous Referee #3 · 7 Sep 2016

General Comments:

This manuscript discusses recent measurements of chlorofluorocarbons CFC-114 and CFC-114a in the atmosphere. The paper fills a gap in the knowledge of these ozone depleting substances concerning the relative abundance of each gas. The paper covers the recent atmosphere as well as atmospheric history going back to the early use of these chemicals, and will be useful in future Ozone Assessment reports. The paper is well-written and comprehensive. I have no problem with the scientific methods or conclusion reached by the authors.

Specific Comments:

Pg. 2, Line 18: Consider using "phase out" in place of "ban". (minor)

Pg. 2, Line 18: Even though you define consumption = production + imports – exports, and the MP does list control in terms of consumption in many cases, it is more general to say that the MP regulates "production and consumption". This would then be consistent with the statement on pg 2, line 26.

Pg. 2, Line 20: Suggest "has started to decline" in place of "started to reduce"

Pg. 5, Line 5: Is this column commercially available? If so, please tell us where you purchased it.

Pg. 5, Line 25: It seems that Laube et al 2010 offers the best description of the dilution method. I think Laube et al 2012 should be Laube et al 2010.

Pg. 6, Line 6: Consider using "mole (mass)" in line 6, since it is used in line 7.

Pg. 7, Line 17: The sentence "The recommended values . . ." is missing something.

Pg. 7, Line 21: I believe that Carpenter and Reimann adopted lifetimes based on SPARC (2013), so it would suffice to use SPARC (2013) as the reference for the 100 and 189 lifetimes.

Pg. 7, Line 5: Not really a complaint, just an observation: You used a relatively sophisticated model, and yet the model is driven largely by data from one site (Cape Grim) or firn air (a smoothed record), and UV absorption data needed to be highly tuned using the parameter F to achieve the recommended total lifetimes. It seems a simple model might suffice given the limited data. I suppose you used a model that was readily available.

Pg. 8, Line 11: Not quire sure what is meant by "early day"

Pg. 8, Line 20: Suggest ". . . ratios below 4%". Given the uncertainties, anything below 4% is clearly not well known.

[Figure]

Pg. 9, Line 10: In, "These two facts imply that increasingly higher emissions of CFC-114a would be needed to sustain increases in mixing ratios above those of CFC-114. " Do you mean that higher emissions of CFC-114a would be required to sustain the observed growth rate of CFC-114a? The mixing ratio of CFC-114a was never higher than that of CFC-114, so I'm not sure what you mean here.

Pg. 9, Line 16: Carpenter and Reimann (2014) state that the assumption of 10% relates to the abundance of CFC-114a relative to CFC-114, based on measurements from 1990. Please provide a reference for the "current" assumption that the emissions fraction is 10%.

Pg. 10 Line 11: Suggest "AFEAS data, which suggests rapidly increasing emissions to more than 5 Gg/year in the late 1940s, are inconsistent with our emissions estimates."

Pg 10, Line 12: Suggest "Emission rates above 5 Gg/yr, suggested by AFEAS, are unlikely to have occurred before the mid-1950's. . ...."

Pg 12, Line 21: I can't find GWP(100)=8490 in Carpenter and Reimann (2014). In the 2014 Ozone Assessment, GWPs were listed in Chapter 5 (Harris and Wuebbles), where I see CFC-114 listed as 8530.

Figure 3 caption: The caption should read: "Global emissions of CFC-114 and CFC-114a derived from Cape Grim observations (solid lines), with uncertainties represented by dashed lines. The dotted lines represent emissions derived purely from firn air data."

Figure S1: I don't know what "scen" means on the titles of the two left-most figures.

---

## Author Comment (AC1) · 9 Nov 2016

We would like to acknowledge the work of the anonymous reviewer and her/his detailed comments which have helped to further improve this manuscript. Below please find responses to all comments.

Referee comment

P5 Calibration – what calibration scale are CFC-114 and CFC-114a reported on?

Author response

As explained in section 3.3 this is a UEA-made calibration scale using our established

volumetric calibration system which minimises differences to gravimetric methods. We have named these and modified the following statements to make this clearer: "Older data had to be transferred to the new calibration scales ("UEA-2014") using repeatedly measured ratios between internal standards. The conversion factor from the old UEA calibration scale (Lee, 1994)..."

Referee comment

P5 L18 Is it reasonable that the average sample precision is 1.1% for both isomers considering one isomer is 14-22x more abundant than the other, and the mole fraction of the samples has increased from 7.9 to 14.8 (CFC-114) and 0.35 to 1.03 (CFC-114a) over time.

Author response

The detection limits of our analytical system are in the sub-ppq range (e.g. Kloss et al., 2014), so the signal-to-noise ratio is not the limiting factor for precisions of either isomer – not even for the deepest firn samples which have as small mixing ratios as 0.06 ppt for CFC-114a.

Referee comment

P5 L26 What is the quoted uncertainty associated with the DuPont sample?

Author response

The sample was provided at >99.8 % purity (weight) as determined by GC-FID and contained small amounts of other trace gases, most notably 0.112 % of CFC-115 and 0.028 % of CFC-13. The latter were successfully removed to below detection limits through transfer into a vacuum-tight canister followed by repeated freezing and evacuating cycles (see statement: "The same dilutions were also analysed in full scan mode to ensure their purity"). This information was added to section 3.3.

In addition our calibration depends heavily on the accuracy of the ratio of CFC-114

and CFC-114a in that sample. DuPont provided no further information with regard to this matter. It is however well known that the molar response factors of isomeric compounds are very similar in Flame Ionisation detectors (e.g. Tong and Karasek, 1984). Nevertheless we have added a respective statement to section 3.3:

"It should also be noted that the accuracy of our calibration is limited by the accuracy of the ratio of CFC-114 and CFC-114a in the sample provided by DuPont, which is unknown. It is however well known that the molar response factors of isomeric compounds are very similar in Flame Ionisation detectors (e.g. Tong and Karasek, 1984), so this is unlikely to be a major limitation of this study."

Referee comment

P5 L31 Use of epoxy resins cover a wide range of potential chemicals and by-products - are the authors sure that no contamination has been introduced to the calibration drums?

Author response

The internal drum surface area exposed to the resin is minimal. In addition the dilution drums were flushed with > 20,000 litres of Nitrogen and no major additional organic compounds were detected in the subsequent blanks. We have added this information to the manuscript and also note that epoxy resin has been successfully used to seal canisters of the whole-air-samplers operated on board the CARIBIC aircraft for more than a decade, including the successful measurement of a large variety of halocarbons and hydrocarbons (Brenninkmeier et al., 2007).

Referee comment

P7 L8 On what evidence is the assumption about latitudinal distribution of CFC-114/114a made?

Author response

The latitudinal distribution is based on the work by McCulloch et al. (1994) which has been added to the manuscript. This emission distribution has been used previously to study the temporal behaviour and global distribution of other long-lived halocarbons (e.g. Reeves et al., 2005, Kloss et al., 2010, Laube at al., 2010, Oram et al., 2012, Newland et al., 2013, Laube et al., 2014). Specifically Reeves et al. (2005) showed that for CFC-11 and CFC-12 the model, with this emission distribution, reproduced southern hemispheric observations to within about 5%.

Referee comment

P7 L17 There appears to be some missing text? "based on work by and".

Author response

Both the sentence "The recommended values mentioned above are based on work by and." and the text above it are intended to convey a similar message. Hence, sentence and text have been revised as follows:

"The rate coefficients of 1.43 x 10-10 cm3 molecule-1 s-1 and 1.62 x 10-10 cm3 molecule-1 s-1 are applied to the reaction of O(1D) with CFC-114 and CFC-114a, based on work by Baasandorj et al. (2013) and Baasandorj et al. (2011), respectively."

Referee comment

P7 L29-32. The quoted uncertainties appear quite small especially 5% for modelling uncertainties. Can the author describe more fully how uncertainties have been derived?

Author response

The modelling uncertainty was estimated to be 5% based on previous work with the model. Generally, the model is able to recreate measurements of long-lived gases at Cape Grim with mainly northern hemisphere emissions and well-established emission histories (e.g., CFC-11, CFC-12) to within 5%.

Referee comment

Figure 2. How can you be sure that the rapid change in ratio of CFC-114a/CFC-114 is not driven by emissions from Asia/Taiwan as detailed in Figure 6?

Author response

The caption of Figure 2 does not claim any reasons for this rapid change. We do provide strong indications in the main text that the ratio change from the 1990s onwards is connected to HFC-134a production. The reviewer is also correct that Figure 6 provides indications for continuing emissions of CFC-114a from East Asia. We do however have no evidence that East Asian HFC production was driving those ratio changes, from the 1990s onwards. This is in fact unlikely as there is and was substantial production of HFC-134a outside of East Asia.

Referee comment

P1 L22 The global ban came into force in 2010 but non-A5 countries in 1996.

Author response

Our statement is correct and has been kept concise for the abstract. The 1996 date is included later in the manuscript.

Referee comment

P1 L37 36 year period

Author response

The period is inclusive of both 1978 and 2014.

Minor suggestions/corrections:

P1 L36 increased from 4.2% to 6.5%

P3 L30 Chan et al., 2006 not 2007

[Figure]

P4 L6 missing a closing bracket ) to close (clean marine ...)

P5 L5 Can you provide the details of the column supplier, diameters, length and film thickness.

P4 L16 Fraser at al., 1986 is not listed in the references?

P14 L30 need full author list not just et al.:

Figure 1. The way the two y-axes are plotted makes the CFC-114 and CFC-114a trends appear to converge between 1980-1990. Can the scales be adjusted to allow the trends and uncertainties to be viewed more easily?

Author response

All changes were made as requested.

---

## Author Comment (AC2) · 9 Nov 2016

Similar to referee #1 we would like to thank this reviewer for her/his work which has led to a further improvement of this manuscript. Below please find responses to all comments.

Referee comment

page 2, line 28: CFC-114a used in HFC-134a production: Could the authors (here or when discussing the Taiwan results) provide information (perhaps through the later mentioned contact at DuPont) on the various production pathways for HFC-134a, which is currently the dominant production method, where (geographically) is HFC-134a produced (presumably mainly China?). These comments apply to here or page 9, line 29 (which pathway is dominant?).

Author response

We recommend Banks et al., 1994 (as cited in the manuscript) for more details on HFC production pathways but consider this level of detail outside the scope of this study. Information on HFC production locations including emissions on an individual country and/or compound basis is not publicly available.

Referee comment

page 3, line 13, this paragraph should be revised as it is somewhat unclear and confusing: Shouldn't Oram (1999) be mentioned here as one of the previous studies on CFC-114/CFC-114a — what did Oram (1999) find out related to the two compounds? Not having access to Lee (1994), p. 3, line 18, it is not clear, if Lee (1994) made measurements of these compounds or if that thesis dealt only with some calibration scales for the compounds under discussion.

Author response

As mentioned in the manuscript the initial calibration of CFC-114 and CFC-114a was explained in the PhD thesis of Lee (1994). We do not use any further data from that thesis and only cite two non-peer-reviewed works (Lee (1994) as well as Oram (1999)) as they provided crucial input to two peer-reviewed publications, both of which are discussed in detail.

Referee comment

line 17 ff. The sentence implies that 'the data compared' were Cape Grim measurements, but wouldn't these be the same data in both cases, those published by Oram (1999) except that i.e. Sturrock et al., 2002 used CFC-114 + CFC-114a while those from UEA (also Oram 1999) separated the two compounds. It would be misleading to call this a 'pure' calibration difference.

Author response

As explained in the manuscript, Sturrock et al. compared CFC-114 measurements from two independent sources: 1) from Antarctic firn air and b) from the Cape Grim archive. Isomers were not separated and are based on an AGAGE calibration scale for the former. We have revised the sentence as follows:

"Their firn air-based data (calibration first reported in Prinn et al. (2000)) were compared with University of East Anglia..."

Referee comment

section 3.2, Analytical technique: could you provide information on potential differences of the molar sensitivities of the two compounds for the two measured mass fragments. Are e.g. for mass 134.96, the peak heights (or areas) per mol of similar size for the two isomers? This information would help to understand potential deficiencies for instrumentation that make combined CFC-114 + CFC114a measurements if reference material differed greatly in composition compared to air samples.

Author response

We thank the reviewer for raising this important point. To address this request we have assessed three measurement days which spanned a large range of mixing ratios measured (CFC-114: 1.08 - 15.85 ppt, CFC-114a: 0.08 -1.09 ppt). On m/z 134.96 CFC-114a had a 2.30 $\pm$ 0.02 larger area response per ppt measured. This does however appear to be a column-specific response factor. A comparison of other CFC isomers (CFC-112, CFC-112a, CFC-113 and CFC-113a) has already revealed that mixing ratios of one isomer could be used to predict those of the other within 23 % on the Agilent GS GasPro column (Kloss et al., 2014). In contrast when determining the molar responses of one of these isomer pairs on the KCl-passivated AlPLOT column (i.e. CFC-113 and CFC-113a as the CFC-112 isomers are not separated) these were very different from expectation i.e. 2.89 instead of 1.68 as expected from the mass

spectra. Nevertheless the mixing ratios of CFC-113 and CFC-113a derived from measurements using the two different columns agree very well. Coming back to the original question we have therefore not included these results in the manuscript but will offer our support to anyone wishing to investigate historical results (e.g. through comparison experiments).

Referee comment

section 3.3 Calibration: pure mixture by DuPont: Could the authors (within confidentiality agreements) provide more details on when (which year) the sample was obtained, if the CFC-114a impurity in CFC-114 production may be constant, which factory it may have been produced (when)?

Author response

The sample was obtained in 2014 but the additional details could unfortunately not be obtained.

Referee comment

page 6 line 1: why is 'of known atmospheric abundance' important? Shouldn't it rather say that a compound was used for which independent calibration exists. Was the comparison with NOAA (2.4%) giving a systematic offset for all 3 samples, and which way, which CFC-12 was higher, the calculated or measured? If the 2.4% is considered to be the accuracy for this CFC-114/a calibration scale, it should be stated clearly (the discussion of uncertainty in a later part of the manuscript suggests so).

Author response

The CFC-12 mixing ratios determined via the UEA system were between 1.6 and 2.4 % higher than the mixing ratios based on the NOAA scale. We have altered the statements to "A CFC for which an independent an internationally recognised calibration exists. . ." and "The three separate calibration analyses were accurate to within 2.4% of NOAA values (using CFC-12 mixing ratios, 2006 NOAA scale) and we therefore

consider the accuracy of the calibration to be 2.4% at the most. As for the calibration precision the $1\sigma$ standard deviation of these calibrations was 1.2 % (CFC-114) and 1.5% (CFC-114a) respectively."

Referee comment

page 6, line 10: One cannot analyze a 'data set' on a GCMS.

Author response

We have altered the statement to: "Both of these data sets originate from measurements on a previous version of the GC-MS system..."

Referee comment

p. 6, line 14: What about the data by Oram (1999), how were these converted? Was Oram (1999) on a preliminary scale? If so, would Lee (1994) mention the difference between scale defined in that thesis and the data published by Oram (1999)? Could the Oram (1999) data be directly converted to the new UEA scale, and if so, what factor, and is it linear. Does UEA plan to give a name to the 'new' scale so to avoid confusion between the past and potential future calibration scales?

Author response

As pointed out above neither of the two works are peer-reviewed literature. To clarify, both Oram (1999) and Lee (1994) provided results based on the same calibration scale.We have changed the statement to "The conversion factor from the old calibration scale (Lee, 1994 and Oram, 1999) as published..." The transfer is linear as can be concluded from the following statement: "To ensure comparability of the data sets, 14 Cape Grim samples collected between 1978 and 2004 have been analysed on both systems and these data agree within uncertainties for both isomers and show no indication for any systematic offset. " The request to name the new scale has already been addressed. Please the respective response to reviewer #1.

Referee comment

page 7, line 14 'at the time'. Why has the model not been re-run with newly-available data? Can the authors assess the error involved with that?

Author response

We have, in the revised version, taken into account the newly published absorption cross section data for CFC-114a by Davis et al. (2016). By re-running the model with the Davis et al. (2016) absorption cross section data, the emissions of CFC-114a have slightly reduced in comparison to the old emissions derived using absorption cross section data by Simon et al. (1988). Also, we have revised the lifetime of CFC-114a from 100 years to 102 years. The latter lifetime is in agreement with the WMO latest recommendation for the lifetime of CFC-12. However, most of the loss of CFC-114a is outside the model domain and so the F value (as explained in the manuscript) has been changed to 0.837, so that the model will give the right lifetime. The range of lifetimes for CFC-114a has also been revised accordingly i.e. from (80 -130 years) to (82 - 133 years). Using the new lifetime and photolysis data has overall caused very small changes to the emission estimates of CFC-114a with equally small knock-on changes to the derived emissions of the sum of the isomers (Figure 4) and the emission ratios displayed in Figure 2. None of these changes alters the overall conclusions of the paper.

Referee comment

page 8, line 4, 'similar relative range'. Could you be more quantitative?

Author response

We have included the numerical values (82 - 133 years) as requested.

Referee comment

page 12, line 21: Arent the GWPs given by Harris et al. (vs Carpenter et al)? I couldn't

find the value of 8490 in either of the two publications.

Author response

We have corrected the statement two-fold: 1) The GWP was reported by Hodnebrog et al., 2013, which has been added to the reference and 2) The number in the text was wrong and has been changed to 8,590 (the calculation was correct).

Referee comment

page 7, line 24: Can you (perhaps in parenthesis) add the numerical value for the height of the model domain?

Author response

The numerical value is 24 km which is already quantified in the manuscript: "The model comprises of grid boxes which have been equally divided into 24 equal-area, zonally-averaged bands and has 12 vertical layers of 2 km depth."

Minor suggestions/corrections:

page 3, line 28, line 30: Years should be in parenthesis, check entire manuscript for this deficiency.

page 7, line 17: incomplete sentence.

page 8, line 3: double mentioning of 'for CFC-114'.

page 9, line 28: wouldn't it be more appropriate to write 'as emissions of the latter continue to increase'.

page 11, line 26: Provide station name and coordinates.

references: several obvious errors, check ms carefully: Baasandorj 2013: space missing; Buizert: 55 Sturges; Marsh: lower case words if a journal article, same for other references (e.g. Laube 2014, Sturges (2012); Subscripted numbers in chemical formulae (Oram 1999), Oram (2012)

Figures: The figures would greatly benefit from red-coloring of the axis label numbers and text for CFC-114a

Supplement: Provide a title for the Supplement including the manuscript title, authors etc.

Supplement: Provide numerical results of all flask measurements and the major calculation results (e.g. yearly emissions and uncertainty bands).

Author response

All changes were made as requested.

———————————————

---

## Author Comment (AC3) · 9 Nov 2016

As an initial remark we would also like to acknowledge the work of this anonymous reviewer which has further improved this manuscript. Below please find responses to all comments.

Referee comment

Pg. 5, Line 25: It seems that Laube et al 2010 offers the best description of the dilution method. I think Laube et al 2012 should be Laube et al 2010.

Author response

The sentence was changed to "Calibration scales were established for CFC-114 and CFC-114a by a two-step dilution process described in Laube et al. (2010) which was improved later (Laube et al., 2012)."

Referee comment

Pg. 7, Line 17: The sentence "The recommended values . . ." is missing something.

Author response

Both the sentence "The recommended values mentioned above are based on work by and." and the text above it are intended to convey a similar message. Hence, sentence and text have been revised as follows:

"The rate coefficients of $1.43 \times 10^{-10}$ cm3 molecule-1 s-1 and $1.62 \times 10^{-10}$ cm3 molecule-1 s-1 are applied for the reaction of O(1D) with CFC-114 and CFC-114a. The recommended values for CFC-114 and CFC-114a mentioned above are based on work by Baasandorj et al., (2013) and Baasandorj et al., (2011), respectively."

Referee comment

Pg. 7, Line 21: I believe that Carpenter and Reimann adopted lifetimes based on SPARC (2013), so it would suffice to use SPARC (2013) as the reference for the 100 and 189 lifetimes.

Author response

CFC-114a was not included in the SPARC report which is why we feel it is necessary to keep both references.

Referee comment

Pg. 7, Line 5: Not really a complaint, just an observation: You used a relatively sophisticated model, and yet the model is driven largely by data from one site (Cape Grim) or firn air (a smoothed record), and UV absorption data needed to be highly tuned

using the parameter F to achieve the recommended total lifetimes. It seems a simple model might suffice given the limited data. I suppose you used a model that was readily available.

Author response

We are using a model that has been proven to work well for a variety of atmospheric trace gases but in particular CFCs.

Referee comment

Pg. 8, Line 11: Not quite sure what is meant by "early day"

Author response

This was one of the first models to infer atmospheric trace gas trends from firn air observations. Many improvements have been made since and we have provided some details in section 3.4

Referee comment

Pg. 9, Line 10: In, "These two facts imply that increasingly higher emissions of CFC-114a would be needed to sustain increases in mixing ratios above those of CFC-114. " Do you mean that higher emissions of CFC-114a would be required to sustain the observed growth rate of CFC-114a? The mixing ratio of CFC-114a was never higher than that of CFC-114, so I'm not sure what you mean here.

Author response

We agree with the referee and have changed the sentence to: "These two facts imply that increasingly higher emissions of CFC-114a would be needed to sustain relative increases (as a percent of the abundance) above those of CFC-114."

Referee comment

Pg. 9, Line 16: Carpenter and Reimann (2014) state that the assumption of 10%

relates to the abundance of CFC-114a relative to CFC-114, based on measurements from 1990. Please provide a reference for the "current" assumption that the emissions fraction is 10%.

Author response

We agree with the referee in that Carpenter and Reimann (2014) state that their assumption is based on data from the 1990s. However, Carpenter and Reimann (2014) apply this assumption to reported current CFC-114 trends and mixing ratios (Table 1-1) and in fact there is no recent evidence in the peer-reviewed literature challenging this assumption.

Referee comment

Pg 12, Line 21: I can't find GWP(100)=8490 in Carpenter and Reimann (2014). In the 2014 Ozone Assessment, GWPs were listed in Chapter 5 (Harris and Wuebbles), where I see CFC-114 listed as 8530.

Author response

We have corrected the statement two-fold: 1) The GWP was reported by Hodnebrog et al., 2013, which has been added to the reference and 2) The number in the text was wrong and has been changed to 8,590 (the calculation was correct).

Referee comment

Figure S1: I don't know what "scen" means on the titles of the two left-most figures.

Author response

The sentence in the figure caption was modified to: "Left: Atmospheric time series corresponding to the modelled firn profiles (black lines, scen: scenario) in comparison to Cape Grim air archive data (blue dots)."

Minor suggestions/corrections:

Pg. 2, Line 18: Consider using "phase out" in place of "ban". (minor)

Pg. 2, Line 18: Even though you define consumption = production + imports – exports, and the MP does list control in terms of consumption in many cases, it is more general to say that the MP regulates "production and consumption". This would then be consistent with the statement on pg 2, line 26.

Pg. 2, Line 20: Suggest "has started to decline" in place of "started to reduce"

Pg. 5, Line 5: Is this column commercially available? If so, please tell us where you purchased it.

Pg. 6, Line 6: Consider using "mole (mass)" in line 6, since it is used in line 7.

Pg. 8, Line 20: Suggest ". . . ratios below 4%". Given the uncertainties, anything below 4% is clearly not well known.

Pg. 10 Line 11: Suggest "AFEAS data, which suggests rapidly increasing emissions to more than 5 Gg/year in the late 1940s, are inconsistent with our emissions estimates."

Pg 10, Line 12: Suggest "Emission rates above 5 Gg/yr, suggested by AFEAS, are unlikely to have occurred before the mid-1950's. . ...."

Figure 3 caption: The caption should read: "Global emissions of CFC-114 and CFC-114a derived from Cape Grim observations (solid lines), with uncertainties represented by dashed lines. The dotted lines represent emissions derived purely from firn air data."

Author response

All changes were made as requested.